# Effect of prenatal care quality on the risk of low birth weight, preterm birth and vertical transmission of HIV, syphilis, and hepatitis

**Debora Melo de Aguiar**[1⊚]*, **Andréia Moreira de Andrade**[1⊚], **Alanderson Alves Ramalho**[1⊚], **Fernanda Andrade Martins**[1⊚], **Rosalina Jorge Koifman**[2⊚], **Simone Perufo Opitz**[1⊚], **Ilce Ferreira da Silva**[2⊚]

1 Postgraduate Program in Public Health, Federal University of Acre, Rio Branco, State of Acre, Brazil,
2 Department of Epidemiology and Quantitative Methods in Health, National School of Public Health, Rio de Janeiro, Rio de Janeiro, Brazil

⊚ These authors contributed equally to this work.
* debora_melo__@hotmail.com

**Data Availability Statement:** All relevant data are available within the paper.

## Abstract

### Background

Averse birth-outcomes still affect newborns worldwide. Although high-quality prenatal care is the main strategy to prevent these outcomes, the effect of prenatal care based on Kotelchuck index combined with consultation contents is still unclear. Thus, this article to evaluate the effect of the quality of prenatal care (PC) process on birth indicators in a cohort of puerperaes who attended maternity hospitals in Brazilian western Amazon, city of Rio Branco, in the state of Acre, Brazil, in 2015.

### Methods

This research was a hospital-based cohort study. The sample consisted of 1,030 women who gave birth in maternity hospitals in the city between April 6 and June 30, 2015. This research was a hospital-based cohort study. The sample consisted of 1,030 women who gave birth in maternity hospitals in Rio Branco between April 6th. and June 30th., 2015. Prenatal care was classified as fully adequate when started ≤4th month; ≥80.0–109% expected consultations for GA according to the Kotelchuck Index; ≥5 records of blood pressure, weight, GA, fundal height, ≥4 records of fetal heart rate, fetal movements or equivalent to 75% of the number of consultations; in addition to recording ABO/RH, hemoglobin, VDRL, urine, glucose, anti-HIV and anti-toxoplamosis during the 1st trimester. The evaluated outcomes were low birth weight (LBW), preterm birth and vertical transmission of human immunodeficiency virus (HIV)/hepatitis/syphilis. Differences between proportions were assessed using the $X^2$ test, and the crude and adjusted odds ratios (OR) (95% CI) were estimated using unconditional logistic regression.

**Funding:** This study was financed in part by Programa de Pesquisa Para o SUS-Fundação de Amparo à Pesquisa do Estado do Acre, PPSUS/FAPAC 2013 6068-14-0000032 and PPSUS/FAPAC 2015 32888.501.20968.22032016. This study was financed in part by the Coordenação de Aperfeiçoamento de Pessoal de Nível Superior-Brasil (CAPES)-Finance Code 001. The funders had no role in study design, data collection and analysis, decision to publish, or preparation of the manuscript.

**Competing interests:** The authors have declared that no competing interests exist.

## Results

Overall cohort, the outcomes incidences were 8.8% for LBW, 9.2% for preterm birth, and 1.1% for vertical transmission (syphilis/HIV/hepatitis). Crude and adjusted OR showed that inadequate PC increased the risk statistically significant of LBW (ORcrude: 1.84; 95%CI: 0.99–3.44; ORadjusted: 1.87; 95%CI: 1.00–3.52), and preterm birth (ORcrude: 1.79; 95% CI: 1.00–3.29; ORadjusted: 3.98; 95%CI: 1.40–11.29).

## Conclusion

The results draw attention to the importance of quality PC in reducing the risks of LBW, preterm birth, and vertical transmission of syphilis/HIV/hepatitis. Moreover, using this proposed quality prenatal care indicator based on Kotelchuck index combined with consultations contents adjusted by GA may accurately predict unfavorable outcomes.

## Introduction

Mortality of children under 5 years of age remains a major public health challenge in low- and middle-income countries, such as Colombia, Mexico, and Brazil, equivalent in Latin America and Caribbean to 16/1000 live births (LB) in 2020 [1]. The neonatal mortality accounting for 46% of all deaths in children under 5 years of age worldwide in 2016 [2]. Brazil has greatly advanced in reducing infant mortality in recent years, whose rates fell from 26/1,000 LB in 1990 [3] to 8/1,000 LB in 2016 [2]. However, these rates vary considerably between the country's regions, in such a way that the infant mortality rates in the northern (22.1/1,000 LB) and northeastern (28.7/1,000 LB) regions remain high [4]. There is evidence in the literature that inadequate prenatal care (PC) may be strongly associated with the high risk of negative outcomes that are well-known risk factors for neonatal death [5].

In the 1990s, Sílvia Takeda observed that the assessment of PC based on objective and measurable criteria is essential for the quality monitoring of the maternal and child health program [6]. Thus, since the 2000s, many authors sought to establish criteria for evaluating the quality of PC based on the parameters of utilization (coverage, gestational age (GA) at the beginning of PC, and number of consultations) [7–10] and content (clinical–obstetric procedures and laboratory tests) [11–14]. Hence, they started using the Kessner index (1973), which is based on the association of GA at the beginning of PC with the number of prenatal consultations [15], and also the Kotelchuck index (1994) [16], which adds to the aforementioned index the adjustment of the number of consultations at any GA.

Although the Kessner index presents limitations regarding accuracy, it is still widely used to assess the utilization of PC, especially in Brazilian studies [12, 17–19]. Although useful, these indices do not consider the essential information regarding the content of consultations (clinical–obstetric procedures and laboratory tests) [20]. Conversely, most of the national [21, 22] and international [23–26] studies that assess the content of PC are limited to evaluating the numbers of each procedure individually [21–26].

In 2012, Anversa et al. proposed the creation of PC adequacy levels based on the association of the Kessner index with the PC content, classifying the evaluation criteria into different levels. Level-1 consisted of the PC utilization according to the Kessner index (GA at first consultation + number of visits); Level-2 consisted of the association between Level-1 and clinical–obstetric procedures; Level-3 consisted of the association between Level-1 and laboratory tests;

and Level-4 consisted of the association between Level-1 and clinical–obstetric and laboratory tests [12]. Therefore, is necessary an adaptation of the levels of the PC evaluation criteria proposed by Anversa et al. (2012) in such a way that quality levels created based on the Kotelchuck index and the number of clinical–obstetric procedures also adjusted for GA. Although there is evidence of greater accuracy in the utilization of the PC adequacy levels proposed by Anversa et al. (2012), the usefulness of this new model in assessing the quality of PC on labor and childbirth outcomes, such as low birth weight (LBW), preterm birth, and vertical transmission of human immunodeficiency virus (HIV)/hepatitis/syphilis, have not yet been evaluated.

Despite the high PC coverage observed in Northern Brazil [9], this region presents one of the poorest performance of at least one VDRL and HIV tests, and the highest preterm-birth, and LBW rates in the country [13]. Such a controversy is partly explained by the PC low-quality program regarding the utilization and contents (clinical-obstetric procedures and laboratory tests). Nevertheless, no study evaluated the effect of PC quality in this region, combining the Kotelchuck index with the PC content adjusted by GA, on birth outcomes. Therefore, the present study aimed to evaluate the PC quality process (utilization and content) effect on birth outcomes in a postpartum women cohort served by maternity hospitals in the city of Rio Branco, state of Acre, Brazil, in 2015.

## Methods

This article is part of two main projects entitled *Utilização de medicamentos durante o período da gestação*, *parto e amamentação em gestantes no município de Rio Branco*, *Acre* ["Use of medicines during pregnancy, childbirth, and breastfeeding in pregnant women in the municipality of Rio Branco, state of Acre, Brazil"] (CAAE: 31007414.0.0000.5010) and *Evolução dos indicadores nutricionais de crianças do nascimento ao primeiro ano de vida em Rio Branco*, *Acre* ["Evolution of nutritional indicators for children from birth to the first year of life in Rio Branco, state of Acre, Brazil"] (CAAE: 40584115.0.0000.5010). Both studies were approved by the Research Ethics Committee of Universidade Federal do Acre [Federal University of Acre], and all participants signed an informed consent form.

Rio Branco is located in the state of Acre, western Brazilian Amazon area, which comprises the greatest ethnic mixed population, and this region it has the greatest miscegenation and presence of indigenous people in the country [27]. Acre encompass around 906.876 inhabitants [28]. Rio Branco city holds only two maternity hospitals, one public and one private that also serves the SUS (Sistema Único de Saúde) in a complementary way.

A hospital-based cohort study was conducted, whose baseline population consisted of all postpartum women living in the urban region of Rio Branco and gave birth in the maternity hospitals of the city between April 6 and June 30, 2015 (in Rio Branco, 95% of childbirths occur in hospitals). The sample consisted of 1,205 postpartum women; 46 (3.8%) of them were excluded because they had multiple pregnancies (n = 11), received PC outside the municipality (n = 26), or did not receive PC (n = 9). Thus, the study population consisted of 1,159 mother–child binomials, of which 129 (10.7%) were lost because they did not have a PC card (n = 6) or did not have information on the number of consultations and/or the time of care onset (n = 123), which are essential for the construction of the Kotelchuck index. The study analyzed 1,030 mother–child binomials, which corresponded to 88.9% of the population eligible for the study. The details of the methodology have been previously published elsewhere [29]. Briefly, interviews for participating in the study were conducted within 48 hours of childbirth using a questionnaire on sociodemographic characteristics and information regarding the current pregnancy. Information on PC was obtained from the pregnant woman's card, while information on labor and childbirth outcomes was obtained from interviews and complemented by data from the medical record.

Maternal and infant outcomes at labor and childbirth outcomes included LBW (babies who are born weighing less than 2,500 grams), preterm birth (birth of a baby at fewer than 37 weeks) and vertical transmission of HIV/hepatitis/syphilis (diagnosed by mother-and-child anti-HIV/hepatitis/syphilis serological tests, retrieved from hospital records). Exposure was defined by combining the utilization criteria (Kotelchuck index) with the PC content (number of obstetric procedures and laboratory tests), as follows:

## PC utilization (time of onset + number of consultations)

Kotelchuck index according to the recommendations of *Programa de Humanização no Pré-natal e Nascimento* [Brazilian Prenatal and Birth Humanization Program] (PHPN) [30]. The Kotelchuck index categories were calculated considering the gestational age (GA) at PN care onset, categorized as 1 and 2 months, 3 and 4 months, 5 and 6 months and 7 to 9 months and the number of consultations expected by the Brazilian Ministry of Health: 6 (37 weeks or more), 5 (33/36 weeks), 4 (29/32 weeks), 3 (28/24 weeks), 2 (<24 weeks), thus the ratio of the number of consultations observed over the number of consultations was obtained by the ratio of the number of consultations observed to those expected for each GA. The final measure is obtained by combining these two dimensions and classifying them into: Adequate plus: beginning before or during the 4th month and 110.0% of consultations ($\geq$7 consultations); Adequate: start before or during the 4th month and 80.0 to 109.0% of consultations (5 or 6 consultations); Intermediate: start before or during the 4th month and 50.0 to 79.0% of consultations (3 or 4 consultations) and Inadequate: start after the 4th month and/or less than 50.0% of consultations (1 or 2 consultations) expected for the gestational age.

## PC content (clinical–obstetric procedures and laboratory tests)

The reference points were defined as follows: for GA, blood pressure, weight, and fundal height, at least five records were considered adequate. Regarding fetal heart rate (FHR) and fetal movement (FM), based on the parameters of the Brazilian Ministry of Health [31], at least four records were considered adequate. These parameters were derived from the studies conducted by Silveira and Santos [11] and later updated in the study by Anversa et al. [12].

## PC utilization + content

PC adequacy levels, such as: Level-1 (Kotelchuck Index), Level-2 (Level-1 + clinical–obstetric procedures), Level-3 (Level-1 + laboratory tests) and Level-4 (Level-1 + clinical–obstetric procedures + laboratory tests).

Information on the utilization and content of each PC consultation was obtained through the PC card, including the place where the PC was received; date of the last menstrual period (LMP), to define the GA for the onset of the PC, preferably, or the ultrasonography (USG); number of PC consultations, record of laboratory tests, weight, blood pressure, fundal height, FHR, and FM. In addition, information regarding socioeconomic, demographic, and current-pregnancy variables was collected through interviews with the women.

Thus, for each PC quality criterion evaluated (Level-1, Level-2, Level-3, and Level-4), the exposed cohort consisted of the set of women who received intermediate/inadequate PC, whereas the nonexposed group consisted of women who received adequate plus/adequate PC.

The distribution of pregnancy, labor and childbirth characteristics according to the criteria of quality levels of PC utilization and content was obtained through absolute and relative frequencies. Differences between proportions were assessed using Pearson's chi-square test for variables with normal distribution and Fisher's exact test for variables with non-normal distribution. A 5% significance level was considered in all analyses.

Subsequently, the crude and adjusted odds ratios (OR) were estimated, with their respective 95% confidence intervals, between the exposure variables: Maternal age (<20 years; 20–34 years; ≥35 years), self-reported skin color (white; mixed-race; others), Education level (elementary school [8 years]; high school [11 years]; college degree [12 years or more]), Marital status (no partner; with partner), Family income, the minimum wage equivalent to R$788 or U$250 in 2015 (up to 1 MW; 1–3 MW; >3 MW), the Brazilian Marketing Research Companies Association (ABEP—Associação Brasileira de Empresas de Pesquisa) classification (A and B; C, D and E), which is responsible for establishing the method of the Economic Classification Criteria in Brazil, Planned pregnancy (no; yes), Prenatal care (public setor; private sector), Abortion (no; yes), Number of childbirths (primiparous; multiparous) and the outcomes, using non-conditional logistic regression, with wald statistical test. We proceeded multivariate analyses, for each association between PC quality care levels (Level-1, 2, 3, and 4) with each outcome (LBW and preterm birth), considering the same set of covariable for adjustment. Thus, multivariate analyzes were proceeded considering the following criteria for covariates inclusion in the model: statistical significance (p<0.20) in the crude OR analysis, biological plausibility in the causal process. The criteria to keep the covariates in the final model included a p-value ≤0.05, biological significance, and a confounding effect detected by modifying the OR magnitude over 10% after adjustment.

All analyses were performed using the SPSS 22.0 statistical package (SPSS Inc., Chicago, United States of America).

## Results

Table 1 shows the distribution of labor and childbirth outcomes outcome indicators according to socioeconomic and demographic characteristics in the study population. According to these data, no statistically significant differences were observed in the frequencies of LBW and vertical transmission (syphilis/HIV/hepatitis). Conversely, frequencies of preterm birth were statistically higher among women who received PC in the public sector (9.9%) when compared with the private sector (4.5%) and among women with a history of abortion (13.0%) when compared with those who have never had an abortion (7.4%).

The highest incidence of LBW was found in women with inadequate PC in all levels of adequacy, equivalent to 9.7% at level-1, 9.0% at level-2, 9.9% at level-3 and 9.7% at level-4. The same result was found for preterm birth, with a higher incidence among women with intermediate/inadequate PC, equivalent to 9.7% at level-1, 11.5% at level-2, 9.9% at level-3 and 10.1% at level-4. The incidence of vertical transmission (08 cases of exposure to syphilis, 02 cases of HIV and 01 case of hepatitis) was also prevalent among women with intermediate/inadequate PC at level-1, level-2, level-3 and level-4, corresponding to 2.2%, 1.2%, 1.3% and 1.1%.

Regarding the PC quality assessment and LBW adjusted for GA (Table 2), women with intermediate/inadequate PC presented the highest incidence of LBW for all levels of assessment of PC quality. Despite a borderline 95% confidence interval, women classified with intermediate/inadequate PC were 62% more likely to have LBW child (95% CI: 0.97–2.72) than those with adequate plus/adequate PC, for level-3 criterion. Similarly, for level-4 criterion, women with intermediate/inadequate PC were 84% more likely to have LBW child (95% CI: 0.99–3.44) than women with adequate plus/adequate PC, despite the borderline 95% CI.

Preterm birth was prevalent among women classified with an intermediate/inadequate PC at all levels of assessment of PC quality. When compared with women who received adequate plus/adequate PC, those with intermediate/inadequate PC presented a higher risk of preterm birth both at Level-2 (OR = 1.89; 95% CI: 1.21–2.97) and despite a borderline 95%CI, in the

**Table 1. Indicators of labor and childbirth outcomes according to socioeconomic and demographic characteristics in a cohort in the municipality of Rio Branco, state of Acre, Brazil (2015).**

| Variable | Low birth weight | | | Preterm birth | | | Vertical transmission | | |
|---|---|---|---|---|---|---|---|---|---|
| | Yes | No | p-value | Yes | No | p-value | Yes | No | p-value |
| **Age (years)** | | | | | | | | | |
| < 20 | 17 (6.9) | 230 (93.1) | | 28 (10.9) | 228 (89.1) | | 4 (1.6) | 252 (98.4) | |
| 20–34 | 61 (9.4) | 589 (90.6) | 0.484 | 60 (9.0) | 604 (91.0) | 0.368 | 7 (1.1) | 657 (98.9) | 0.410 |
| ≥35 | 10 (9.3) | 97(90.7) | | 7 (6.4) | 103 (93.6) | | 0 (0.0) | 110 (100) | |
| **Skin color** | | | | | | | | | |
| White | 8 (7.4) | 100 (92.6) | | 6 (5.5) | 103 (94.5) | | 1 (0.9) | 108 (99.1) | |
| Mixed-race | 70 (8.4) | 761 (91.6) | 0.140 | 84 (9.8) | 769 (90.2) | 0.289 | 9 (1.1) | 844 (98.9) | 0.938 |
| Others | 10 (15.4) | 55 (84.6) | | 5 (7.4) | 63 (92.6) | | 1 (1.5) | 67 (98.5) | |
| **Education level** | | | | | | | | | |
| Elementary school | 27 (10.5) | 230 (89.5) | | 26 (9.8) | 240 (90.2) | | 4 (1.5) | 262 (98.5) | |
| High school | 47(9.0) | 476 (91.0) | 0.249 | 54 (10.1) | 479 (89.9) | 0.262 | 4 (0.8) | 529 (99.2) | 0.576 |
| College degree | 14(6.3) | 210 (93.8) | | 15 (6.5) | 216 (93.5) | | 3 (1.3) | 228 (98.7) | |
| **Marital status** | | | | | | | | | |
| No partner | 12 (7.7) | 144 (92.3) | 0.606 | 19 (11.8) | 142 (882) | 0.218 | 2 (1.2) | 159 (98.8) | 0.815 |
| With partner | 76 (9.0) | 772 (91.0) | | 76 (8.7) | 793 (91.3) | | 9 (1.0) | 860 (99.0) | |
| **Family income** | | | | | | | | | |
| Up to 1 MW | 11 (8.0) | 126 (92.0) | | 15 (10.6) | 127 (89.4) | | 2 (1.4) | 140 (98.6) | |
| 1–3 MW | 46 (9.3) | 451 (90.7) | 0.531 | 49 (9.7) | 457 (90.3) | 0.339 | 4 (0.8) | 502 (99.2) | 0.527 |
| >3 MW | 26 (11.4) | 203 (88.6) | | 16 (6.8) | 221 (93.2) | | 4 (1.7) | 233 (98.3) | |
| **ABEP class** | | | | | | | | | |
| A and B | 15 (76) | 182 (92.4) | 0.557 | 16 (8.0) | 185 (92.0) | 0.453 | 3 (1.5) | 198 (98.5) | 0.462 |
| C, D, and E | 71 (8.9) | 724 (91.1) | | 79 (9.7) | 737 (90.3) | | 8 (1.0) | 808 (99.0) | |
| **Planned pregnancy** | | | | | | | | | |
| No | 59 (9.4) | 567 (90.6) | 0 373 | 65 (10.1) | 578 (89.9) | 0.177 | 5 (0.8) | 638 (99.2) | 0.233 |
| Yes | 29 (7.8) | 344 (92.2) | | 29 (7.6) | 353 (92.4) | | 6 (1.6) | 376 (98.4) | |
| **Prenatal care** | | | | | | | | | |
| Public sector | 77 (8.8) | 796 (91.2) | 0 873 | 89 (9.9) | 807 (90.1) | **0.042** | 10 (1.1) | 886 (98.9) | 0.698 |
| Private sector | 11 (84) | 120 (91.6) | | 6 (4.5) | 128 (95.5) | | 1 (0.7) | 133 (99.3) | |
| **Abortion**** | | | | | | | | | |
| No | 45 (10.7) | 375 (89.3) | 0 315 | 32 (7.4) | 401 (92.6) | **0.022** | 2 (0.5) | 431 (99.5) | 0.336 |
| Yes | 16 (7.8) | 189 (92.2) | | 27 (13.0) | 181 (87.0) | | 3 (1.4) | 205 (98.6) | |
| **Number of childbirths** | | | | | | | | | |
| Primiparous | 31 (7.2) | 402 (92.8) | 0 117 | 44 (9.9) | 399 (90.1) | 0.495 | 6 (1.4) | 437 (98.6) | 0.544 |
| Multiparous | 57 (10.0) | 514 (90.0) | | 51 (8.7) | 536 (91.3) | | 5 (0.9) | 582 (99.1) | |

Source: The Authors.

Level-4, women with intermediate/inadequate PC presented 79% higher risk of preterm birth than women with adequate PC quality (OR = 1.79; 95% CI: 1.00–3.29) (Table 2).

Vertical transmission of syphilis/HIV/hepatitis was more frequent among women in the intermediate/inadequate category when compared with those in the adequate plus/adequate PC at all levels of PC quality. However, only in the Level-1 PC quality criterion (Kotelchuck index) women classified in the intermediate/inadequate category presented a statistically higher risk of vertical transmission of syphilis/HIV/hepatitis (OR = 3.49; 95% CI: 1.06–11.52) than those classified as receiving adequate plus/adequate PC (Table 2).

**Table 2. Indicators of labor and childbirth outcomes according to the Kotelchuck's utilization criteria and associated with the assessment of quality of prenatal care content.**

| PC adequacy levels | Low birth weight | | Preterm birth | | Vertical transmission (syphilis/HIV/hepatitis) | |
|---|---|---|---|---|---|---|
| | Incidence (8.8%) | OR (95% CI) | Incidence (9.2%) | OR (95% CI) | Incidence (1.1%) | OR (95% CI) |
| LEVEL-1 (KOTELCHUCK INDEX—NUMBER OF CONSULTATIONS AND TIME OF ONSET) | | | | | | |
| Adequate plus/adequate | 63 (8.5%) | 1 | 69 (9.0%) | 1 | 5 (0.7%) | 1 |
| Intermediate/inadequate | 25 (9.7%) | 1.16 (0.71–1.88) | 26 (9.7%) | 1.09 (0.68–1.74) | 6 (2.2%) | 3.49 (1.06–11.52) |
| LEVEL-2 (KOTELCHUCK INDEX + OBSTETRIC PROCEDURES) | | | | | | |
| Adequate plus/adequate | 39 (8.5%) | 1 | 30 (6.4%) | 1 | 4 (0.9%) | 1 |
| Intermediate/inadequate | 49 (9.0%) | 1.06 (0.68–1.64) | 65 (11.5%) | 1.89 (1.21–2.97) | 7 (1.2%) | 1.45 (0.42–4.90) |
| LEVEL-3 (KOTELCHUCK INDEX + LABORATORY TESTS) | | | | | | |
| Adequate plus/adequate | 20 (6.3%) | 1 | 25 (7.8%) | 1 | 2 (0.6%) | 1 |
| Intermediate/inadequate | 68 (9.9%) | 1.62 (0.97–2.72) | 70 (9.9%) | 1.30 (0.81–2.09) | 9 (1.3%) | 2.05 (0.44–9.55) |
| LEVEL-4 (KOTELCHUCK INDEX + OBSTETRIC PROCEDURES + LABORATORY TESTS) | | | | | | |
| Adequate plus/adequate | 12 (5.5%) | 1 | 13 (5.9%) | 1 | 2 (0.9%) | 1 |
| Intermediate/inadequate | 76 (9.7%) | 1.84 (0.99–3.44) | 82 (10.1%) | 1.79 (1.00–3.29) | 9 (1.1%) | 1.23 (0.26–5.71) |

Source: The Authors.

In the multivariate analysis (Table 3), it was observed that at Level-3 of PC quality (Kotelchuck index + laboratory tests), women with intermediate/inadequate PC were at a 71.0% (95% CI: 1.01–2.91) higher risk of LBW when compared with those with adequate plus/adequate PC adjusted for the ABEP socioeconomic classes. Conversely, at Level-4 of PC quality (Kotelchuck index + clinical-obstetric procedures + laboratory tests), women with intermediate/inadequate PC presented an 87.0% (95% CI: 1.00–3.52) higher risk of LBW than those with adequate plus/adequate PC, even after adjusting for age and skin color.

When compared with women with adequate plus/adequate PC at Level-2 (Kotelchuck index + clinical–obstetric procedures) and Level-3 (Kotelchuck index + laboratory tests) of PC assessment, the risk of preterm birth was 2.42 (95% CI: 1.31–4.47) and 2.21 (95% CI: 1.08–4.52) times higher, respectively, among women with intermediate/inadequate PC. When considering Level-4 of PC assessment (Kotelchuck index + clinical–obstetric procedures + laboratory tests), women with intermediate/inadequate PC presented 3.98 times higher risk of preterm birth (95% CI: 1.40–11.29) when compared with those receiving adequate plus/adequate PC, regardless of whether they had a partner and history of abortion.

## Discussion

The results that the incidence of LBW in women with PC Adequate plus/adequate decreased from the moment the Kotelchuck index + laboratory procedures were considered (6.3%) or when all recommendations at Level-4 were considered (5.5%). Thus demonstrating the importance of all procedures, especially laboratory procedures to reduce conditions that cause LBW. Preterm birth had a lower incidence in women with Adequate plus/adequate prenatal care considered the Kotelchuck index + obstetric procedures (6.4%) and all procedures at Level-4 (5.9%). Demonstrating the importance of such procedures, especially obstetrics to reduce prematurity.

Vertical transmission (syphilis/HIV/hepatitis) had the lowest incidence among women with adequate plus/adequate prenatal care, considering the Kotelchuck + laboratory tests index (0.6) and only the Kotelchuck index (0.7%). Demonstrating the importance to early

**Table 3. Association between prenatal care quality assessment levels and indicators of labor and childbirth outcomes according to socioeconomic and demographic characteristics, Rio Branco, state of Acre, Brazil (2015).**

| PC adequacy levels | Low birth weight | | | | Preterm birth | | | |
|---|---|---|---|---|---|---|---|---|
| | Level-1 | Level-2 | Level-3 | Level-4 | Level-1 | Level-2 | Level-3 | Level-4 |
| Adequate plus/ adequate | 1 | 1 | 1 | 1 | 1 | 1 | 1 | 1 |
| Intermediate/ inadequate | 1.18 (0.72–1.93) | 1.05 (0.67–1.63) | **1.71 (1.01–2.91)** | **1.87 (1.00–3.52)** | 1.29 (0.72–2.31) | **2.42 (1.31–4.47)** | **2.21 (1.08–4.52)** | **3.98 (1.40–11.29)** |
| **Age (years)** | | | | | | | | |
| <20 | 1 | 1 | - | 1 | - | - | - | - |
| 20–34 | 1.45 (0.83–2.55) | 1.45 (0.83–2.50) | - | 1.42 (0.81–2.49) | - | - | - | - |
| ≥35 | 1.47 (0.65–3.36) | 1.47 (0.65–3.33) | - | 1.50 (0.66–3.43) | - | - | - | - |
| **Skin color** | | | | | | | | |
| White | 1 | 1 | - | 1 | - | 1 | 1 | |
| Mixed-race | 1.16 (0.54–2.49) | 1.16 (0.54–2.48) | - | 1.22 (0.57–2.63) | - | 2.62 (0.62–11.20) | 2.97 (0.70–12.60) | |
| Others | 2.31 (0.86–6.21) | 2.29 (0.85–6.16) | - | 2.48 (0.92–6.68) | - | 1.05 (0.14–7.92) | 1.32 (0.18–9.83) | |
| **Marital status** | | | | | | | | |
| No partner | - | - | - | - | - | - | - | 1.20 (0.58–1.48) |
| With partner | - | - | - | - | - | - | - | 1 |
| **ABEP class** | | | | | | | | |
| A and B | - | - | 1 | - | - | - | - | - |
| C, D, and E | - | - | 1.20 (0.67–2.16) | - | - | - | - | - |
| **Planned pregnancy** | | | | | | | | |
| No | - | 1.26 (0.79–2.00) | - | - | - | - | - | - |
| Yes | - | 1 | | | | | | |
| **Prenatal care** | | | | | | | | |
| Public sector | - | - | - | - | 1.74 (0.60–5.02) | - | - | - |
| Private sector | - | - | - | - | 1 | - | - | - |
| **Abortion** | | | | | | | | |
| No | - | - | - | - | 1 | 1 | 1 | 1 |
| Yes | - | - | - | - | **1.91 (1.11–3.30)** | **1.89 (1.09–3.27)** | **2.04 (1.18–3.54)** | **2.04 (1.18–3.53)** |

prenatal care, in addition to the minimum number of consultations to detect morbidities that can be treated with the aim of preventing vertical transmission.

Accordingly, it was observed that pregnant women with PC classified as intermediate/inadequate as per the utilization criteria of the Kotelchuck index and content (clinical–obstetric procedures and laboratory tests) presented higher risks of unfavorable outcomes such as LBW, preterm birth, and vertical transmission of syphilis/HIV/hepatitis. These findings corroborate the results of previous studies conducted both in high-income countries, such as Canada [24], USA [32], and Belgium [33], and in low and middle-income countries, such as Ghana [34] and Zimbabwe [35].

It is important to emphasize that improving the quality of care in low and middle-income countries is still a challenge and inequalities in health-care quality have not been systematically

examined. A recent study Demographic and Health Surveys (2007–2016) and Multiple Indicator Cluster Surveys in 91 low-income and middle-income country (LMIC), antenatal care quality lagged behind antenatal care coverage the most in low-income countries, where 86.6% of women accessed care but only 53.8% (44.3–63.3) reported blood pressure monitoring, urine and blood testing [36].

Zhou et al. (2019), acknowledging the issue of LBW and the importance of evaluating the content of PC consultations in assessing the quality of the program, conducted a cross-sectional study in 42 counties in China. Among the 5,891 newborns weighed at birth, 6.6% had LBW. The beginning of the PC in the first trimester did not show a statistically significant relationship with LBW. However, no record of maternal blood pressure (ORadj = 1.39; 95% CI: 1.04–1.86), no record of blood tests (ORadj = 1.42; 95% CI: 1.09–1.84), and no record of urinalysis (ORadj = 1.39; 95% CI: 1.08–1.80) during the consultations increased the risk of LBW [37]. These findings suggest that the content of PC plays a significant role in the risk of unfavorable maternal and child outcomes such as LBW. Hence, it would no longer be possible to develop indicators for assessing the quality of PC without considering the effects of the PC content (laboratory tests and obstetric procedures).

However, a significant association was found when considering at least one content criterion after adjusting for sociodemographic variables. These variables are important for identifying possible complications and reducing outcomes, such as preterm birth, which are considered as the main causes of neonatal mortality worldwide. In Rio Branco, an incidence of preterm births was observed among women with inadequate PC (7.5%), which is similar to that high-income countries such as Canada (7.5%) [24] and Brussels (7.2%) [33]. In the retrospective cohort study conducted in the capital of Belgium in 2008, no statistically significant association was found between preterm birth and adequacy of the number of consultations according to the Kotelchuck index (ORadj = 3.13; 95% CI: 0.36–27.04). However, corroborating the findings of the present study, when considering the content of the consultations (two ultrasounds, one in the 1st trimester and one in the 2nd trimester; at least six blood pressure measurements, one in the 1st trimester, two in the 2nd trimester, and three in the 3rd trimester; and two complete blood counts, one in the 1st trimester and one in the 3rd trimester of pregnancy), a four-times lower risk of preterm birth (ORadj = 0.21; 95% CI: 0.06–0.68) was found among women with PC classified as adequate when compared with those classified as receiving inadequate PC [33].

Regarding the vertical transmission of syphilis/HIV/hepatitis, in 2015, the year in which the survey was conducted, Acre was the state with the 5th highest number of cases of gestational syphilis and the municipality of Rio Branco presented a rate of vertical transmission of syphilis equal to 5.8/100,000 LB [38]. In Ohio (USA), there was an increase in the rate of congenital syphilis from 2/100,000 in 2003 to 9.3/100,000 in 2016 [39]. On the other hand, in the state of Santa Catarina (Brazil), there was an increase in the vertical transmission of syphilis from 0.5% in 2007 to 5.4/1,000 LB in 2015, with an increase of 0.9% between 2007 and 2017 [40]. The literature considers socioeconomic barriers, especially factors related to access and quality of PC, to be associated with the increased risk of vertical transmission of syphilis [39, 40].

Although substantial progress has been made in increasing access to health services in low and middle-income countries, the quality of care provided across different countries and health conditions remains low, thus, it is estimated that more than half of pregnancies in women with syphilis result in an adverse outcome without adequate treatment [41].

In a cross-sectional study conducted in Rio Branco between 2007 and 2015, it was observed that 64.3% of notifications of HIV-positive pregnant women on the Brazilian Information System on Notifiable Diseases (SINAN) occurred in the 3rd trimester of pregnancy [41]. The late onset of PC (after the 1st trimester) could delay the early examination for detection.

Furthermore, for women who already know that they have the disease, the late onset may result in infection of the fetus due to the duration of exposure to a high viral load. In addition, the lack of information on the PC card would compromise chemoprophylactic actions at the time of childbirth, such as the administration of antiretroviral therapy to the newborn in the first 24 hours of birth [42]. Thus, the relationship between low quality of PC and the risk of vertical transmission of diseases has been confirmed both for syphilis and for other infectious diseases whose vertical transmission can be prevented using chemoprophylaxis, such as HIV and hepatitis B and C [40–42].

Considering the financial difficulties of low and middle-income countries, a group of researchers from Pelotas emphasize the importance of an indicator that can measure beyond prenatal coverage and that is designed so that these countries have the necessary information for their construction and easy to use, noting that the quality of care in health provided across in low-income and middle-income countries remains low and hinders progress in improving health outcomes [43].

Therefore, we recognize the importance of establishing an evaluation model that can be built in low and middle-income countries [44]. However, we considered that is necessary to verify the register or not of other essential procedures in addition to blood pressure measured, blood sample and urine sample collected to identify negative outcomes such as LBW and pre-term birth that can cause neonatal death and prevent the occurrence and achieve the goals established by Sustainable Development Goals (2030).

Among the advantages of the present study, the authors wish to highlight the methodological rigor with the purpose of reducing possible information bias. Data were collected directly from the PC card by the study researchers, which allowed the construction of the exposure variable (PC quality based on utilization and content). Another advantage concerns the adjustment for GA at childbirth, both for information on the PC and on the outcome variables, such as LBW and preterm birth. These factors permitted the accurate and precise classification of both the exposure (PC quality) and the outcomes that could be affected by GA. In addition, the present study was performed within a cohort of LB from the only two maternity hospitals in Rio Branco. Thus, the findings of the present study hold generalization potential for the entire urban population of Rio Branco (state of Acre).

Conversely, some limitations of the present study must also be considered. As this is a proposal for a new prenatal assessment indicator associating the Kotelchuck index with the content (laboratory tests and clinical–obstetric procedures) adjusted for GA, comparisons with studies that used the indicators of utilization and content were made separately. Furthermore, in 2015 (the year in which the study was conducted), the criteria established by *Rede Cegonha* (strategy implemented by the Brazilian Ministry of Health aimed at improving the health care of pregnant women) in 2011 were not yet fully implemented in Rio Branco. Hence, the criteria used for the construction of the utilization and content indicators were those established by the standards of the Brazilian Prenatal and Birth Humanization Program (PHPN). Although this methodological decision may limit comparisons with studies that were conducted in locations where the recommendations of *Rede Cegonha* (2011) were already in force, the proposal for the association of utilization and content indicators (adjusted for GA) can be perfectly adapted insofar as the recommendations and regulations are updated over time.

Lastly, although we have also intended to assess the effect of the PC quality on the syphilis, HIV and hepatitis vertical transmission, the few number of cases found in the studied cohort limited the multiple analysis for this outcome. Thus, studies with greater sample size are necessary to analyze the effect of PC quality on syphilis, HIV and hepatitis vertical transmission in Rio Branco, Acre, since the incidence of this outcome was very low in this area.

## Conclusion

The results of present study leaded us to conclude that despite the low HIV/Syphilis/Hepatitis vertical transmission incidence observed in Rio Branco, the LBW and preterm birth incidence were higher in pregnant women with PC classified as intermediate/inadequate than high-income countries such as Canada, Brussels and Belgium. Additionally, the proposed method for assessing PC quality process by combining utilization with the content adjusted by gestational age, was able to predict the effect of intermediate/inadequate PC on LBW risk, even adjusting by maternal age and skin color. Also, the risk of preterm birth among those with intermediate/inadequate PC was almost 4-folds higher than those with adequate plus/adequate PC, even adjusted by marital status, and abortions.

In this sense, reducing the inequalities in low-income countries is necessary, as well as reinforce the importance of high-quality prenatal care and of using an accurate indicator for PC quality assessment that combine utilization and content adjusted by GA. Therefore, adequate and adjusted methods for evaluating the quality of services offered are essential to identify and overcome the flaws/gaps, and their impact on reducing birth bad outcomes.

## Author Contributions

**Conceptualization:** Debora Melo de Aguiar, Andréia Moreira de Andrade, Rosalina Jorge Koifman, Ilce Ferreira da Silva.

**Data curation:** Debora Melo de Aguiar, Alanderson Alves Ramalho.

**Formal analysis:** Debora Melo de Aguiar, Alanderson Alves Ramalho, Ilce Ferreira da Silva.

**Investigation:** Fernanda Andrade Martins.

**Methodology:** Debora Melo de Aguiar, Andréia Moreira de Andrade, Rosalina Jorge Koifman, Ilce Ferreira da Silva.

**Project administration:** Alanderson Alves Ramalho.

**Resources:** Fernanda Andrade Martins, Rosalina Jorge Koifman, Simone Perufo Opitz.

**Software:** Alanderson Alves Ramalho.

**Supervision:** Andréia Moreira de Andrade, Rosalina Jorge Koifman, Ilce Ferreira da Silva.

**Visualization:** Andréia Moreira de Andrade, Fernanda Andrade Martins, Simone Perufo Opitz.

**Writing – original draft:** Debora Melo de Aguiar, Simone Perufo Opitz, Ilce Ferreira da Silva.

**Writing – review & editing:** Debora Melo de Aguiar, Andréia Moreira de Andrade, Alanderson Alves Ramalho, Fernanda Andrade Martins, Rosalina Jorge Koifman, Ilce Ferreira da Silva.

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
