## [Decision Letter · Decision Letter 0]

6 Jul 2022

PGPH-D-22-00381

Effect of prenatal care quality on the risk of low birth weight, preterm birth and vertical transmission of HIV, syphilis, and hepatitis.

Dear Dr. Aguiar,

Thank you for submitting your manuscript to PLOS Global Public Health. After careful consideration, we feel that it has merit but does not fully meet PLOS Global Public Health’s publication criteria as it currently stands. Therefore, we invite you to submit a revised version of the manuscript that addresses the points raised during the review process.

We look forward to receiving your revised manuscript.

Kind regards,

Julia Robinson

Executive Editor

Journal Requirements:

a. Please clarify all sources of funding (financial or material support) for your study. List the grants (with grant number) or organizations (with url) that supported your study, including funding received from your institution. 

b. State the initials, alongside each funding source, of each author to receive each grant.

c. State what role the funders took in the study. If the funders had no role in your study, please state: “The funders had no role in study design, data collection and analysis, decision to publish, or preparation of the manuscript.”

d. If any authors received a salary from any of your funders, please state which authors and which funders.

Additional Editor Comments (if provided):

Reviewers' comments:

Reviewer's Responses to Questions

**Comments to the Author**

1. Does this manuscript meet PLOS Global Public Health’s publication criteria? Is the manuscript technically sound, and do the data support the conclusions? The manuscript must describe methodologically and ethically rigorous research with conclusions that are appropriately drawn based on the data presented.

Reviewer #1: Partly

Reviewer #2: Partly

2. Has the statistical analysis been performed appropriately and rigorously?

Reviewer #1: No

Reviewer #2: No

3. Have the authors made all data underlying the findings in their manuscript fully available (please refer to the Data Availability Statement at the start of the manuscript PDF file)?

Reviewer #1: No

Reviewer #2: Yes

4. Is the manuscript presented in an intelligible fashion and written in standard English?

Reviewer #1: No

Reviewer #2: Yes

5. Review Comments to the Author

Reviewer #1: Thanks for the work you've done. It's has important findings, especially for poor resource settings in a large country, like Brazil. Despite its importance, your work would benefit from some amendments in the text. Please, consider my comments below.

Abstract

Line 29: I think is obsolete and prejudiced the usage of the terms “developed and developing countries”. Please, consider my suggestion of replacing it by high-income and low and middle-income countries. Also, there’s a typo in the same line.

Line 31: Maybe I’m outdated, but the common term used is “postpartum women” instead of “puerperaes”. But this is out of my scope in this review.

Line 32: The Methods section needs some rearrangement. I would say that the definition of adequate PC is essential to understand the studied groups, instead of describing who is exposed or not. Covariates used to adjust the models were not described.

Line 33: Same as the comment above, the Results section also needs some changes. I’d like to see the crude and adjusted figures for OR.

Introduction

The comment about the terms “developed and developing countries” and “puerperae” applies to the whole text.

Line 52: I don’t think that is plausible to compare mortality rates between Latin American countries, where progress has been reached in the last decades on reducing the burden of under 5-y mortality, with Bangladesh that still struggles to reduce the high mortality rates.

Methods

Please, provide details about the study site. Where is Rio Branco located, how many people live there, is it a big city, how many maternity wards exist in the city, among other basic characteristics?

Line 135: What’s the difference between “outcomes at childbirth and birth”?

Line 136: It’s important to provide the definitions used for each outcome. Additionally, it would be great to see the details about the Kotelchuck index, because I think a bit though to keep reading the findings of your research without further details.

It’s crucial to provide information about the covariates available and selected to perform the multivariate analysis.

Results

Line 187: Usually, the first thing to describe in this section is the overall frequencies/prevalence of the exposures and outcomes. Table 1 contains this information by levels of exposure.

Line 189: Findings for LBW were not statistically significant for any exposure level investigated. I think it does not worth mentioning in detail such findings as they are given in Table 1.

Line 209: Table 2 should be the first table of your paper.

Line 217: I found confusing the description of adjusted models as different covariates were used in each model. Please, provide information in the Methods section of the covariates considered in the analysis. Did you select a different set of covariates for each level of exposure? I miss that part.

Table 3: It looks unfinished. The style is different from the other tables.

Discussion

Lin 245: This first paragraph looks like the reasons for conducting your study and would be better placed in the Introduction. The first paragraph of the Discussion is a synthesis of the main findings of your research. Please, consider rewriting it.

Overall: The authors seemed to focus on seeking other findings in the literature that corroborated with theirs, instead of deeply discussing your results in light of the challenges and strategies to overcome such barriers for optimal prenatal care, especially in poor-resource settings in Brazil. My recommendation is to review this section thoroughly.

Recently, a new analysis was published about the quality and content of prenatal care in low- and middle-income countries. I suggest you take a look at this work which might be useful.

https://jogh.org/documents/2021/jogh-11-04008.pdf

https://equityhealthj.biomedcentral.com/articles/10.1186/s12939-021-01440-3

Reviewer #2: Dear editor,

Thanks for the opportunity to review this manuscript.

Methods

1) Variables that were considered for adjustment should be described in methods (paragraph starting at line 174).

Results

1) Odds ratio for LBW was similar regardless the PC level, since confidence intervals included 1 in the four analysis conducted. However, authors suggest that they were different in text (line 187). Consider rewriting it.

2) The same occurs with the incidence of preterm birth; only women with inadequate level 2 were at higher risk for preterm birth, but not those in group 4.

3) Many cells of table 3 are empty, so I cannot evaluate it.

Conclusions are not based on the results authors provided.

6. PLOS authors have the option to publish the peer review history of their article (what does this mean?). If published, this will include your full peer review and any attached files.

**Do you want your identity to be public for this peer review?** For information about this choice, including consent withdrawal, please see our Privacy Policy.

Reviewer #1: No

Reviewer #2: **Yes: **Jose Paulo Guida

---

## [Decision Letter · Decision Letter 1]

6 Nov 2022

PGPH-D-22-00381R1

Effect of prenatal care quality on the risk of low birth weight, preterm birth and vertical transmission of HIV, syphilis, and hepatitis.

Dear Dr. Aguiar,

Thank you for submitting your manuscript to PLOS Global Public Health. After careful consideration, we feel that it has merit but does not fully meet PLOS Global Public Health’s publication criteria as it currently stands. Therefore, we invite you to submit a revised version of the manuscript that addresses the points raised during the review process.

We look forward to receiving your revised manuscript.

Kind regards,

Melissa Morgan Medvedev, M.D., Ph.D.

Academic Editor

Journal Requirements:

Additional Editor Comments (if provided):

Reviewers' comments:

Reviewer's Responses to Questions

**Comments to the Author**

1. If the authors have adequately addressed your comments raised in a previous round of review and you feel that this manuscript is now acceptable for publication, you may indicate that here to bypass the “Comments to the Author” section, enter your conflict of interest statement in the “Confidential to Editor” section, and submit your "Accept" recommendation.

Reviewer #1: All comments have been addressed

Reviewer #3: (No Response)

2. Does this manuscript meet PLOS Global Public Health’s publication criteria? Is the manuscript technically sound, and do the data support the conclusions? The manuscript must describe methodologically and ethically rigorous research with conclusions that are appropriately drawn based on the data presented.

Reviewer #1: Partly

Reviewer #3: Yes

3. Has the statistical analysis been performed appropriately and rigorously?

Reviewer #1: I don't know

Reviewer #3: Yes

4. Have the authors made all data underlying the findings in their manuscript fully available (please refer to the Data Availability Statement at the start of the manuscript PDF file)?

Reviewer #1: No

Reviewer #3: No

5. Is the manuscript presented in an intelligible fashion and written in standard English?

Reviewer #1: No

Reviewer #3: Yes

6. Review Comments to the Author

Reviewer #1: Thank you for your thorough review and for accepting most of my comments. The work has improved.

I still have some other minor comments, displayed below:

Line 35: “Thus, this article to evaluate the”. There’s a missing word here, and could it be: ‘this article aims to evaluate’. Or any other that fits here.

Line 43: “>80.0 - 109% expected consultations for GA according to the Kotelchuck Index”. Didn’t understand these figures. What 109% represents?

Line 49: You stated ‘level 4’ of adequacy. Maybe you could give information on the other levels or simply omit this information.

Lines 55-59: Please, just give information for findings statistically significant.

Line 64-66: I think it’s important to provide an average rate for under 5y mortality for these countries.

Line 134: What SUS means? I think this is the first time it appears on the text.

Lines 156-57: Were the serological tests performed by the study team or information was retrieved from hospital records? If performed by the study team, you must provide details about the procedures for each test.

Lines 169-173: By reading the description of the method, it looks like that if a women attended more than the expected number of antenatal care appointments, she would be classified as over 110% of the necessary appointments. Is that right? I still think that the description of the method is a bit tough to follow. Would be possible to convert the % into absolute numbers, like instead of using 110%, use XX ANC appointments?

Line 205: Please, inform how many years each educational category corresponds.

Line 206: What’s de minimum wage? In Brazilian currency and US dollars by the time the data was collected.

Line 207: What’s ABEP class and what it classifies? Remember, you’re submitting your work to an international journal and most readers are not familiar with the Brazilian system.

My overall comment is about English writing. I strongly recommend you to use certified services of English translation/verification.

Reviewer #3: Review of the manuscript “Effect of prenatal care quality on the risk of low birth weight, preterm birth and vertical transmission of HIV, syphilis, and hepatitis”.

This is a secondary data study from two prospective studies aiming to evaluate the effect of the quality of the prenatal care process on birth indicators in a cohort of mothers who attended maternity hospitals in the city of Rio Branco in the state of Acre, Brazil, in 2015.

Abstract/Title

1) It is well known that prenatal care quality is associated with worst outcomes for the baby. What is new and interesting in this study are the indicators for prenatal care quality and their ability to predict bad outcomes. The title and the abstract should be based on that strength.

2) Second sentence in the abstract has a typo: “… to evaluate to evaluate…”

3) The first sentence in the results (abstract) is not clear: “The incidences were 8.8% for LBW, 9.2% for preterm birth (FOR THE ENTIRE COHORT?), 39.4% and 1,1% (???) for vertical transmission…”

4) The conclusion in the text is much better than in the abstract (again… to conclude that quality o prenatal care is associated with worst outcomes is a weak conclusion.

Introduction

5) The introduction is too long. I suggest reducing at least one paragraph.

6) The objective at the end of the introductions should reflect the conclusion at the end of the manuscript (as explained before)

Method

7) The study is well design. I would just include what inadequate, intermediate, adequate, adequate plus means in terms of number of consultations since that´s an important part of the study.

Results

8) Is reasonable to group syphilis, HIV and hepatitis as they are not frequent, but it would be interesting to know how many of each did you have in your cohort.

9) Table 3 should also include vertical transmission (or just exclude this outcome of the study).

Discussion

10) Again, the authors should better discuss the association of the indicators with vertical transmission and why they are different than the other outcomes (or just exclude vertical transmission of the outcome).

7. PLOS authors have the option to publish the peer review history of their article (what does this mean?). If published, this will include your full peer review and any attached files.

**Do you want your identity to be public for this peer review?** For information about this choice, including consent withdrawal, please see our Privacy Policy.

Reviewer #1: No

Reviewer #3: No

---

## [Decision Letter · Decision Letter 2]

22 Feb 2023

Effect of prenatal care quality on the risk of low birth weight, preterm birth and vertical transmission of HIV, syphilis, and hepatitis.

PGPH-D-22-00381R2

Dear Mrs Aguiar,

We are pleased to inform you that your manuscript 'Effect of prenatal care quality on the risk of low birth weight, preterm birth and vertical transmission of HIV, syphilis, and hepatitis.' has been provisionally accepted for publication in PLOS Global Public Health.

Best regards,

Julia Robinson

Executive Editor

Reviewer Comments (if any, and for reference):

Reviewer's Responses to Questions

**Comments to the Author**

1. If the authors have adequately addressed your comments raised in a previous round of review and you feel that this manuscript is now acceptable for publication, you may indicate that here to bypass the “Comments to the Author” section, enter your conflict of interest statement in the “Confidential to Editor” section, and submit your "Accept" recommendation.

Reviewer #1: All comments have been addressed

2. Does this manuscript meet PLOS Global Public Health’s publication criteria? Is the manuscript technically sound, and do the data support the conclusions? The manuscript must describe methodologically and ethically rigorous research with conclusions that are appropriately drawn based on the data presented.

Reviewer #1: Partly

3. Has the statistical analysis been performed appropriately and rigorously?

Reviewer #1: I don't know

4. Have the authors made all data underlying the findings in their manuscript fully available (please refer to the Data Availability Statement at the start of the manuscript PDF file)?

Reviewer #1: No

5. Is the manuscript presented in an intelligible fashion and written in standard English?

Reviewer #1: No

6. Review Comments to the Author

Reviewer #1: Nothing to ad. All my comments were properly addressed.

7. PLOS authors have the option to publish the peer review history of their article (what does this mean?). If published, this will include your full peer review and any attached files.

**Do you want your identity to be public for this peer review?** For information about this choice, including consent withdrawal, please see our Privacy Policy.

Reviewer #1: No
